# Single Particle Automated Raman Trapping Analysis

Jelle Penders[1,2,3], Isaac J. Pence[1,2,3], Conor C. Horgan[1,2,3], Mads S. Bergholt[1,2,3], Christopher S. Wood [1,2,3], Adrian Najer[1,2,3], Ulrike Kauscher[1,2,3], Anika Nagelkerke[1,2,3] & Molly M. Stevens [1,2,3]

Enabling concurrent, high throughput analysis of single nanoparticles would greatly increase the capacity to study size, composition and inter and intra particle population variance with applications in a wide range of fields from polymer science to drug delivery. Here, we present a comprehensive platform for Single Particle Automated Raman Trapping Analysis (SPARTA) able to integrally analyse nanoparticles ranging from synthetic polymer particles to liposomes without any modification. With the developed highly controlled automated trapping process, single nanoparticles are analysed with high throughput and sensitivity to resolve particle mixtures, obtain detailed compositional spectra of complex particles, track sequential functionalisations, derive particle sizes and monitor the dynamics of click reactions occurring on the nanoparticle surface. The SPARTA platform opens up a wide range of new avenues for nanoparticle research through label-free integral high-throughput single particle analysis, overcoming key limitations in sensitivity and specificity of existing bulk analysis methods.

[1] Department of Materials, Imperial College London, London SW7 2AZ, UK. [2] Department of Bioengineering, Imperial College London, London SW7 2AZ, UK. [3] Institute of Biomedical Engineering, Imperial College London, London SW7 2AZ, UK. These authors contributed equally: Jelle Penders, Isaac J. Pence. Correspondence and requests for materials should be addressed to M.M.S. (email: m.stevens@imperial.ac.uk)

The analysis of nanoparticles in solution is a crucial step for a wide range of research fields commonly including polymer particles and vesicles for drug delivery systems, such as liposomes and polymersomes. Particle sizing and compositional analysis are typically achieved by combining a range of laser-based diffraction and spectroscopic techniques. Dynamic Light Scattering (DLS) and Nanoparticle Tracking Analysis (NTA) are generally employed to determine the particle population size distribution[1–3], whereas compositional analysis can be conducted using Mass Spectrometry (MS) techniques as well as (Fourier-transform)-Infrared (IR) spectroscopy[4–6] among others, depending on the type and size of particles. The reliance on multiple techniques for sizing and composition analysis brings the drawback that these methods vary in sample requirements such as concentration, preparation, and sensitivity. For nanoparticles in particular, population heterogeneity can severely affect their function and applicability, which cannot be resolved with these conventional bulk analysis techniques[7,8].

Here, we introduce a comprehensive nanoparticle analysis platform based on Raman spectroscopy to provide simultaneous size and composition analysis on a single particle basis. Raman spectroscopy is a well-established characterisation technique that can provide label-free compositional data based on inelastic scattering of incident laser photons and has been applied for samples ranging from simple powders to cells, when using complex 3D imaging[9]. The obtained Raman spectrum gives a molecular fingerprint of the chemical constituents of the sample. To interrogate individual particles without confounding contributions of substrates, Raman spectroscopy can be applied in combination with optical trapping. Pioneered by Ashkin[10,11], a particle can be levitated or trapped due to the radiation pressure created by the laser focus. Nanoparticles in the Rayleigh limit ($r \ll \lambda$) are trapped due to a difference in the polarisability of the particle compared to the solution, leading to a dipole gradient force. This force scales with laser intensity and decreases with increasing distance from the focal volume, which directs the particle into the optical trap at the focal point of the laser[12]. For Raman spectroscopy this is ideal, as the laser creating the particle trap can simultaneously be used to acquire a Raman spectrum of the particle. This has sparked a wide range of studies investigating various micro- and nano-sized particles such as microdroplets[13,14] and silicon nanoparticles[15]. Of particular interest is the use of Raman spectroscopy to analyse the composition and heterogeneity of vesicular structures posed for drug delivery systems, such as liposomes[12,16] and polymersomes[17,18]. These particles can be made from a wide range of amphiphilic molecules, to obtain vesicles with a variety of compositions, size ranges and physical properties[16,19]. It has previously been shown that Raman spectroscopy can be used successfully to analyse the composition of polymersomes a few micrometres in size[20]. However, despite recent advances, the use of Raman spectroscopy for single particle analysis suffers from a major limitation, namely the fact that the particles need to be manually trapped inside the laser or lifted from a substrate[21,22]. This significantly limits the number of particles that can be analysed as the process is both slow and labour-intensive. The very limited particle throughput also obstructs any investigation of composition heterogeneity with sufficient statistical power.

In addition to particle compositional analysis, prior studies have shown the possibility of investigating dynamic events or reactions occurring on the particle surface by monitoring the spectrum of optically-trapped particles over an extended time. Examples include observation of polymerisation reactions[23], solid phase particle assisted peptide synthesis[24] or measurements of analyte concentrations in liposomes[25]. Further analysis of trapped particles includes the investigation of the influence of optical trapping forces on micrometre-sized liposomes through the addition of a solution marker such as perchlorate ions[12]. In addition, perchlorate ions have been employed as an internal standard to measure surrounding solute concentrations[25] and have been shown to be impermeable to lipid membranes[26].

To address the imperative need for large scale integral size and composition characterisation of single nanoparticles, we developed a novel platform for Single Particle Automated Raman Trapping Analysis (SPARTA). SPARTA enables high throughput, routine analysis of individual nanoparticles in solution without any need for particle labelling or modification. Here, we demonstrate a thorough analysis of the composition of liposome and polymersome systems, as well as the ability to resolve mixtures and investigate particle heterogeneity using the SPARTA platform. In addition to particle compositional analysis, we show that the SPARTA platform is ideally suited for monitoring sequential functionalisation of polystyrene nanoparticles, as well as tracking the dynamics of a click reaction on the particle surface. Lastly, by taking advantage of the high-throughput measurement capability, we demonstrate that perchlorate addition can be used in a radical new way, to allow single particle sizing of the trapped particles. SPARTA opens up a plethora of exciting new applications to analyse inter and intra sample heterogeneity, complex mixtures, on-line reaction monitoring and integrated simultaneous sizing of single particles in high detail.

## Results

**SPARTA system design and validation.** The SPARTA system is based on a high-end confocal Raman spectroscopy set-up where the laser, camera and spectroscope are simultaneously controlled via custom, in-house MATLAB scripts, for automated single particle trapping and Raman spectral acquisition. To enable the application of SPARTA for comprehensive particle analysis, we developed three distinct modes of operation (Fig. 1).

The first mode comprises the functionalisation and composition analysis (Fig. 1-I) by acquisition of high quality Raman spectra for single particles in solution, allowing investigation of their composition and verification of the presence of specific functionalisations. The SPARTA system has the key advantage of enabling automated analysis of hundreds of particles, compared to existing systems only capable of analysing in the range of tens of particles. The automation and up scaling of the number of particles analysed enables a means to analyse particle variance both on a single particle basis and at population level for complex mixtures of particles.

The second mode of the SPARTA platform (Fig. 1-II) is solution marker mediated sizing analysis. By combining the high throughput single particle analysis with a perchlorate ion standard, we demonstrate here that the size of the particle in the trap can be estimated simultaneously when acquiring its compositional information. A particle entering the trap displaces its volume of perchlorate ion solution from the trap. By measuring the decrease in perchlorate signal in the Raman spectrum, relative to trapping particles of known size, a calibration curve can be obtained to relate the perchlorate signal in the spectrum to the size of the particle in the trap, provided it is smaller than the confocal volume. This enables particle sizing on a single particle basis with the simultaneous collection of compositional data, allowing direct acquisition and correlation of particle size and composition, where hitherto a combination of several analysis techniques had to be used and size and composition could not be compared on a per particle basis.

The third mode of the SPARTA platform is on-line dynamic reaction monitoring (Fig. 1-III); tracking the progress of a

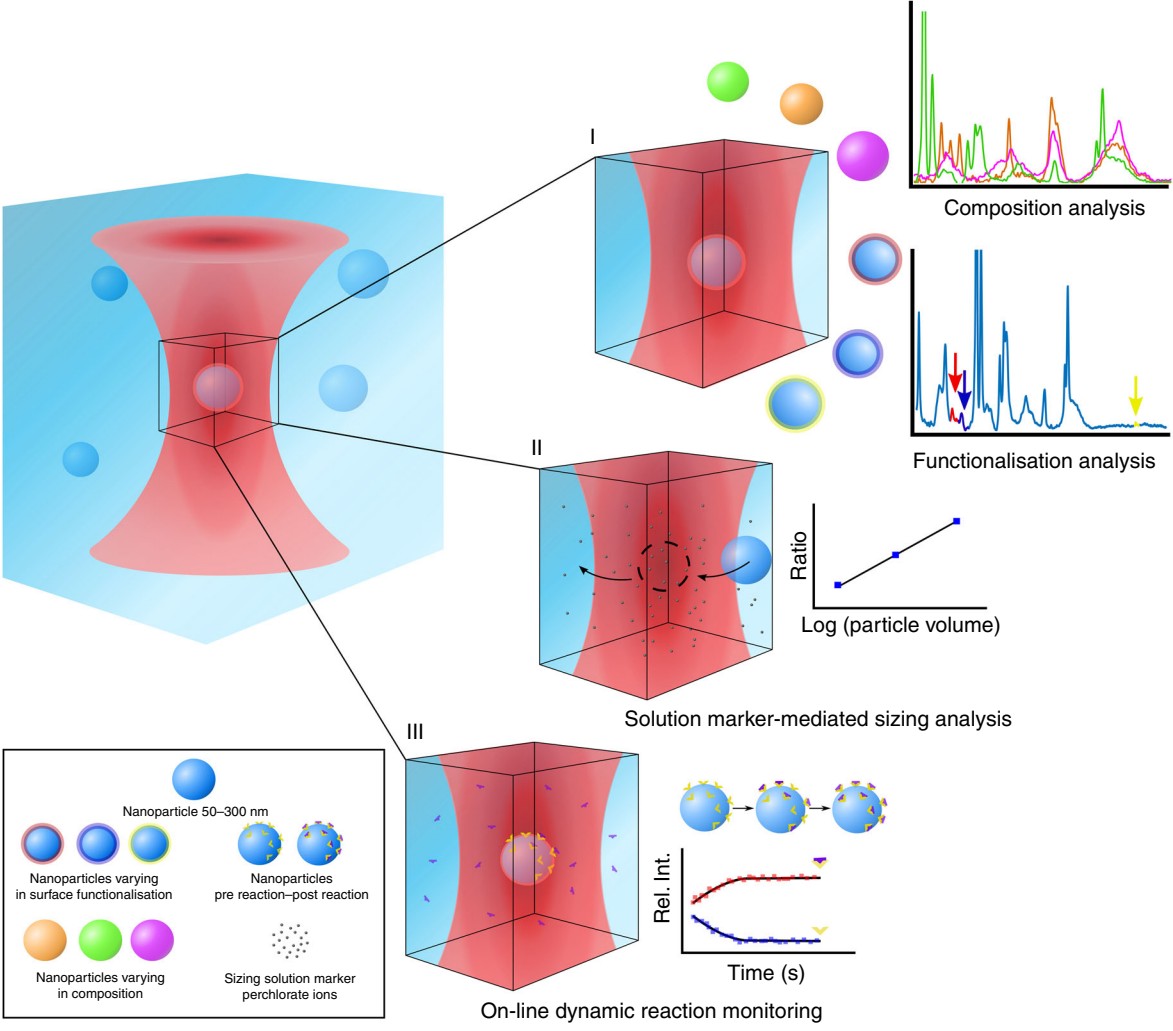

**Fig. 1** Schematic overview of the functionalities of the SPARTA platform. The capacity to trap nanoparticles in the size range of 50–300 nm automatically and with a high degree of control allows the analysis of the particles in three distinct, complementary modes: (I) the first mode is functionalisation and composition analysis, which provides the Raman spectrum of single particles in solution to verify functionalisation and provide detailed analysis of their composition. This also enables the analysis of the heterogeneity of single particles within a uniform population or complex mixtures. (II) The second mode is solution marker mediated sizing analysis, allowing the estimation of the size of a trapped particle by measuring the relative displacement of perchlorate ions from the confocal volume. (III) The third mode is on-line dynamic reaction monitoring, which allows the tracking of the progress of a reaction occurring on the surface of the particle, either on a single particle for the duration of the reaction or a different single particle at distinct time points

dynamic event on single particles. This can be achieved by either holding a single particle in the trap for the duration of the reaction, or by trapping a new particle at subsequent time points to compare reaction progress on a particle by particle basis. The single particle sampling aspect of this technique permits investigation of reaction kinetics, identifying if the reaction occurs simultaneously at the same rate on each particle, or on specific single particles at any one time, as would be the case for reactions limited by catalyst availability. These results can further be correlated to bulk dynamics, which can be tracked with conventional methods.

To facilitate the aforementioned areas of application, extensive control is required over the trapping process, detailed in the SPARTA process flow (Fig. 2a). The core concept of the automated trapping process is the alternation between short acquisitions, called iterations, and longer acquisitions for high signal to noise ratio (SNR) spectra. Prior to spectral acquisition, the user can define several parameters (Fig. 2a-I) through the software interface, including the number of acquisitions ($n_a$), and the times for iteration, high SNR acquisition and laser disabling

between each trap. To allow for automated trap recognition, an initial particle is trapped and a characteristic peak in the spectrum is selected, along with its median height as a threshold (Fig. 2a-II). This results in the recognition of a successful trap if the chosen peak is above the threshold intensity during an iteration acquisition. Next, the acquisition iteration loop is initiated (Fig. 2a-III, IV). If the thresholding peak is not above threshold intensity in the iteration spectrum (Fig. 2a-IV), a maximum of 10 iterations is performed. If the peak signal fails to exceed the selected threshold during any of these iterations, the trap is registered as unsuccessful and the laser is momentarily disabled. When a successful particle trap is recognised (Fig. 2a-V), a longer acquisition is taken to obtain a high SNR spectrum of the trapped particle (Fig. 2a-VI). Lastly, the trap is momentarily disabled by turning off the laser to allow the particle to diffuse away. This iteration process allows for a much higher turnover of trapping attempts and acquisitions, since it only permits the acquisition of longer, high SNR spectra when trapping is successful. The automated determination of a successful particle trap obviates the need to record data from iterations without sufficient signal

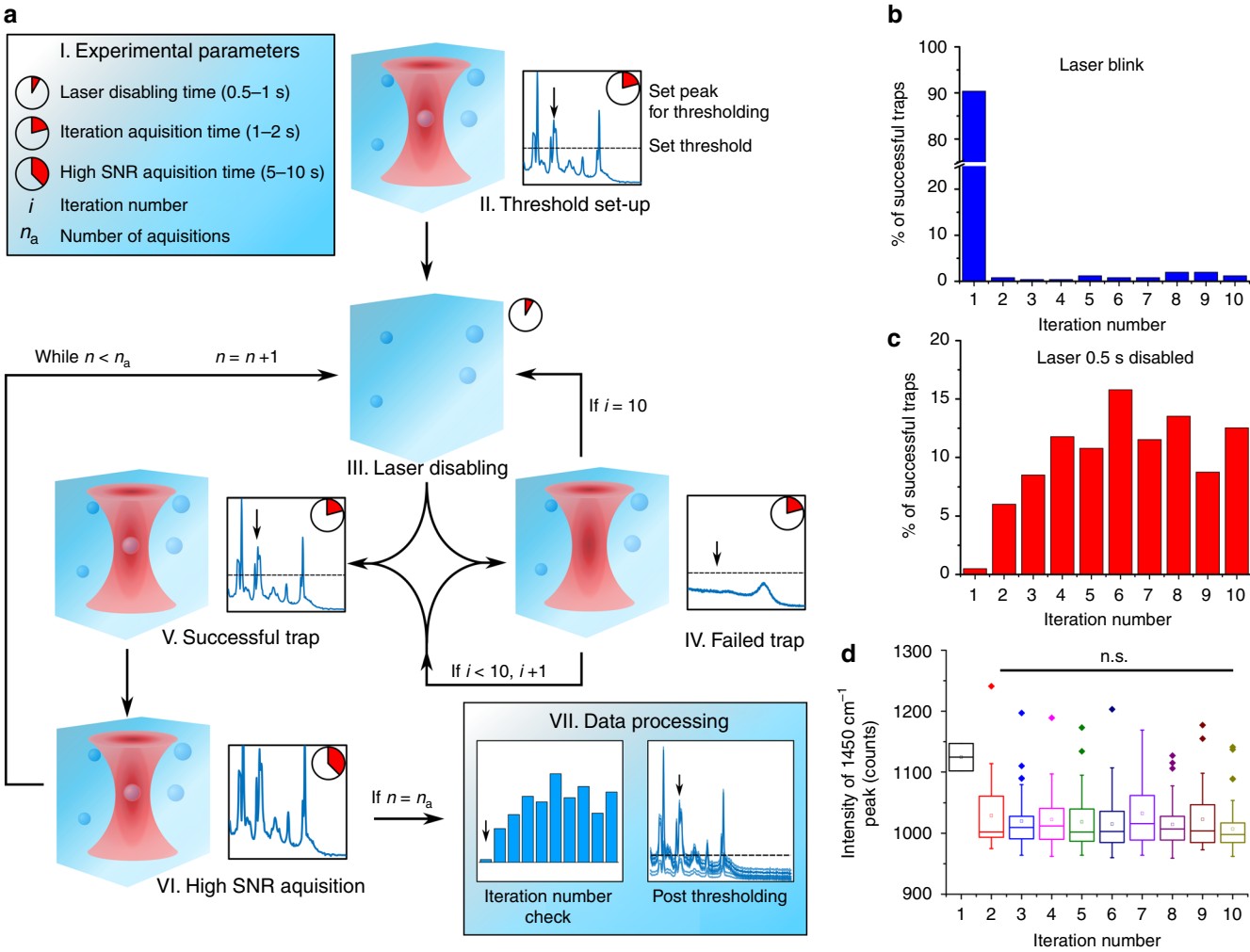

**Fig. 2** Overview of the SPARTA process flow and trapping control validation. **a** Process flow starting with the set-up of the acquisition parameters (I) and thresholding based on the signal obtained from an initial particle trap to set the peak position and height used to determine a successful particle trap (II). Next, the acquisition loop is initiated (III, IV) where quick scans are iterated until a trap is recognized (V) after which a high SNR acquisition is taken and saved (VI). The laser is disabled (III) to allow the particle to diffuse away and the process is repeated. After $n_a$ acquisitions the data can be processed (VII) for further analysis. **b** Iteration number distribution when rapidly blinking the laser after each trap. **c** Iteration number distribution when turning the laser off for 0.5 s. **d** Box and whisker plot showing the signal intensity distribution for a characteristic peak versus the iteration number of the trap, no significant influence was found between any of the iteration numbers >1. One way ANOVA, with Holm-corrected multiple comparisons test, $n = 355$ combined traps. Indicated are the median (horizontal line inside the box), mean (□), minimum and maximum values as the ends of the whiskers and outliers (♦)

relative to the threshold as can be seen by the comparison of the spectral lineshape for a successful polystyrene particle trap (Fig. 2a-V) versus a trapping iteration containing only the fluid medium (Fig. 2a-IV). For each successful particle trap, the acquisition parameters, time and iteration number are recorded in a trapping log file alongside the high SNR spectral data, to be used for verification and post processing (Fig. 2a-VII). The iteration number is used to verify an adequate laser disabling time in an iteration number check. The spectra can also be put through post thresholding to pick out any 'false positive' traps if the thresholding intensity was initially set too low. Conventional Raman spectral processing can be performed afterwards, including cosmic spike removal, baseline subtraction and normalisation.

An adequate laser disabling time ensures momentary disengaging of the trap to allow the particle to diffuse away and is essential for avoiding duplicate analysis of the particle. If the disabling time is too short (Fig. 2b), e.g. only blinking the laser, it results in more than 90% of the successful traps occurring at the first iteration, thus with a high likelihood that the same particle is repeatedly trapped before it can diffuse away. However, a laser

disabling time between 0.5 and 1 s is sufficient to ensure <1% trapping at the first iteration number (Fig. 2c), which is in agreement with diffusion speeds reported in literature[27]. In addition, it was verified that the iteration number at which a spectrum is acquired does not affect the peak intensity of the spectrum, for instance due to fluctuations in the laser power. As shown in (Fig. 2d) for a characteristic peak at 1450 cm$^{-1}$ ($CH_2$ vibration), the peak intensity is not significantly altered by the iteration number of a successful trap, with the notable exception of particles trapped at iteration 1. While it has been previously shown that NIR lasers do not induce significant photodamage on trapped particles[14,27,28], we have further verified this by trapping and holding a 1,2-dipalmitoyl-sn-glycero-3-phosphocholine (DPPC) liposome for 5 min in the laser and taking spectra at regular intervals. The standard deviation of the mean spectrum is low and does not show appreciable peak changes indicative of photodamage (Supplementary Fig. 1a), nor does the surface plot indicate spectral changes over time (Supplementary Fig. 1b) apart from small overall intensity changes attributable to the slight fluctuations in laser power.

**Functionalisation and composition analysis.** The primary mode of operation for the SPARTA system is single particle functionalisation and composition analysis. Here, either particles of varying composition or particles with the same core composition but varying surface functionalisations are analysed. This was demonstrated by trapping DPPC liposomes and liposomes containing 50% deuterated DPPC (d-DPPC) with respect to their DPPC content, referred to as d-DPPC liposomes. In the Raman spectra of the two samples (Fig. 3a) a clear C-D signal can be observed around 2105 cm$^{-1}$ in the spectrum for the d-DPPC liposomes, which is absent for the DPPC liposomes. Due to the high-throughput automated operation, mixtures of particles can be analysed and resolved. To demonstrate this, a 50–50 v/v % mixture of DPPC liposomes and d-DPPC liposomes was made. With SPARTA, hundreds of particles were trapped and the spectra were analysed by a Gaussian mixture analysis based on the histogram of the intensity of the C-D vibration at 2105 cm$^{-1}$. This resulted in a clear bimodal distribution of the histogram (Fig. 3b) covering 44 and 56% of the traps for DPPC and d-DPPC respectively, showing that the mixture can be clearly resolved. Alternatively, the mixture can be resolved by cluster analysis, as can be seen in the Ward's dendogram (Fig. 3c), showing two main clusters of spectra relating to the non-deuterated and deuterated populations. A small deuterium signal was observed in the non-deuterated classed spectra, which possibly resulted from lipid exchange between deuterated and non-deuterated liposomes. As deuterium containing molecules are very strong Raman scatterers[29], only a small percentage of deuterated molecules are required to generate a detectable Raman signal.

With SPARTA, more subtle differences in composition can also be detected, which is of high relevance in nanoparticle analysis, particularly in the field of nanomedicine, where an exact definition of the composition of nanomaterials is of the utmost importance, yet often elusive. To demonstrate this, we made two formulations of polymersomes, one from poly(2-methyloxazoline-b-dimethylsiloxane-b-2-methyloxazoline) (PMOXA-b-PDMS-b-PMOXA) denoted further as ABA and the other of ABA supplemented with 25 wt.% PDMS-b-heparin, prepared as described by Najer et al.[30], and termed ABA-heparin polymersomes. SPARTA analysis of the particles (Fig. 3d) showed characteristic peaks that can be attributed to PDMS (Supplementary Table 1) for both preparations. In the spectra obtained from the ABA-heparin polymersomes, additional peaks can be seen and assigned to the saccharide units of heparin, as indicated by the arrows in the insert. In addition, the average area under the curve for the normalised spectra of the PDMS peaks at 490 and 709 cm$^{-1}$ (Supplementary Table 1) is lower for the ABA-heparin polymersomes, with a ratio compared to the ABA polymersomes of 1:0.898. This is in excellent agreement with the theoretical ratio of 1:0.843, corresponding to 83 and 70 wt.% PDMS respectively for the ABA and ABA-heparin polymersomes, as calculated from the molecular weights and quantities added of the PMOXA-b-PDMS-b-PMOXA and PDMS-b-heparin block copolymers. Next, we made a 50–50 v/v % mixture of ABA and ABA-heparin polymersomes and analysed it with SPARTA. These mixtures can be resolved by either an unsupervised classification such as principal component analysis (PCA) or a supervised method such as partial least squares discriminant analysis (PLSDA) where the data from the pure populations is used to build a model and subsequently applied to classify the spectra obtained from the particles in the mixture. A 2 component PCA model (Fig. 3e) shows clear distinction into two clusters, mainly based on

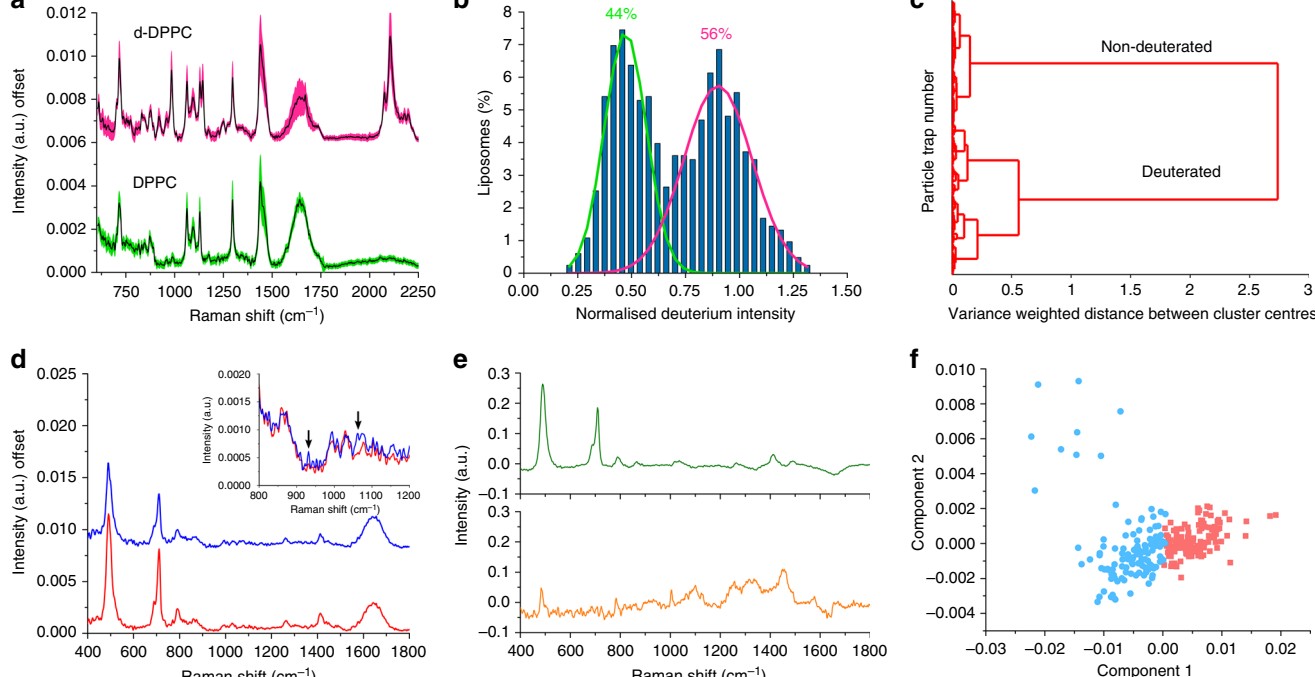

**Fig. 3** SPARTA composition analysis of liposomes and polymersomes. **a** Offset averaged Raman spectra of DPPC liposomes (green mean ± s.d., $n = 649$) and 50% d-DPPC liposomes (magenta mean ± s.d., $n = 340$). **b** Gaussian 2 component mixture analysis based on expectation maximisation; bimodal histogram of the CCD counts at the C-D peak position (2105 cm$^{-1}$) of a 50–50 v/v % mixture of DPPC and d-DPPC liposomes ($n = 828$) with 44% classed as DPPC and 56% as d-DPPC. **c** Ward's clustering dendogram of a 50–50 v/v % mixture of DPPC and d-DPPC liposomes ($n = 828$). **d** Offset averaged Raman spectra of ABA polymersomes (red, $n = 99$) and ABA-heparin polymersomes (blue, $n = 98$), insert showing a close-up of the region with arrows denoting heparin signals around 930 and 1070 cm$^{-1}$. **e** 2 component PCA decomposition analysis of a 50–50 v/v % mixture of ABA and ABA-heparin polymersomes ($n = 263$), component 1 (top) and component 2 (bottom). **f** PCA scores post Ward's clustering (light red ABA-like, light blue ABA-heparin-like)

variance in PDMS composition. With PCA based Ward's clustering, 49% of the spectra were classified as ABA-like polymersomes, 47% as ABA-heparin-like polymersomes and 4% were not classified as belonging to either of those clusters (Fig. 3f). PLSDA classified 38% of the acquired spectra as ABA polymersomes and 62% as ABA-heparin polymersomes (Supplementary Fig. 2), based on a model of pure particles that achieved 100% sensitivity and 97.9% specificity for ABA polymersomes by venetian blinds cross validation (10 splits). To verify that either type of polymersome was trapped in a random manner, without time dependence, the scores for PCA component 1 were plotted against the trap number, resulting in a random distribution of polymersome classes over time, proving no bias over time for either composition (Supplementary Fig. 3).

In addition to vesicular systems, the composition and sizing of a wide variety of solid nanoparticles is critically important in a range of applications from drug delivery to catalysis[31,32]. To validate the use of the SPARTA platform for analysis and verification of successful nanoparticle functionalisation, we devised a model system based on the sequential functionalisation of polystyrene (PS) nanoparticles via disulphide exchange (Fig. 4a). Commercially obtained amine functionalised PS particles with an average size of 200 nm were sulfhydryl functionalised by addition of 2-iminothiolane. After purification by centrifugation and resuspension, we analysed the particles with the SPARTA platform which verified the presence of sulfhydryl, as seen from the signals in the spectra arising from the *S-H* bend vibration around 936 cm$^{-1}$ (Supplementary Table 1) (black, Fig. 4b). Next, we added an excess of 5,5′-dithio-bis-(2-nitrobenzoic acid) (DTNB) which reacted with the sulfhydryl functionalised particles to form disulphide bonds. After further purification, the Raman spectra (magenta, Fig. 4b) showed the clear presence of disulphide bonds. These bonds are dynamic covalent bonds and can thus be exchanged upon addition of another moiety containing a sulfhydryl functionality. We demonstrated this by addition of a tripeptide, consisting of cysteine and two tyrosines (CYY), as peptide functionalisation of nanoparticles is a desirable and widely used strategy, especially in the field of drug delivery[33,34]. The cysteine residue provides a sulfhydryl functionality and the tyrosines exhibit a characteristic Raman peak, due to the aromatic *C = C* bonds. Upon addition of the peptide to the purified particles, the solution turned

characteristically yellow, demonstrating that the 2-nitro-5-thiobenzoate dianion (TNB$^{2-}$) was formed. This indicated that disulphide exchange had occurred and confirmed the specific conjugation of the tripeptide to the particle. After purification, we acquired Raman spectra of the particles, showing the presence of characteristic tyrosine peaks (blue, Fig. 4b). To complete the cycle, Tris(2-carboxyethyl) phosphine (TCEP) can be added to either disulphide-containing particle solution to recover the sulfhydryl functionalisation (Fig. 4c). At each step the size distribution of the particles was measured by DLS, verifying particle stability during the sequential functionalisation and purification (Supplementary Fig. 4).

**Solution marker mediated sizing analysis.** The second mode of the SPARTA system allows for simultaneous estimation of the particle size inside the trap, alongside the acquisition of a high SNR compositional spectrum. As illustrated (Fig. 1b), a particle entering the trap displaces the same volume as itself out of the surrounding solution from the confocal volume, leading to a decrease in the perchlorate signal in the measured spectrum. Perchlorate is particularly suitable for this application as it has a single sharp Raman peak around 938 cm$^{-1}$ (Supplementary Fig. 5). As the PS signal also increases upon increasing particle size inside the confocal volume, the perchlorate signal is best quantified by a ratio contribution to the spectrum according to: Perchlorate ratio $= \left(A_t - A_p\right)A_p^{-1}$ where $A_t$ is the total area under the curve of the spectrum and $A_p$ the area for the perchlorate peak. The ratio will increase when $A_p$ approaches zero (particle completely fills the trap) and go to zero (no particle) once $A_p$ approaches $A_t$.

As a model system, we analysed PS particles of 200, 100 and 50 nm with the SPARTA platform upon addition of 50 mM sodium perchlorate to the solution. We found that the perchlorate ratio was distinct between the various sizes and are within bandwidths with minimal overlap (Fig. 5a). The average ratios can be plotted against the log particle volume (calculated from the size provided by the manufacturer) resulting in an excellent linear correlation ($R^2 = 0.99$) (Fig. 5b). The linear fit of the calibration curve can be used to estimate the particle size of the individual particles trapped with the SPARTA system. These sizes were binned identically to the corresponding DLS number distribu-

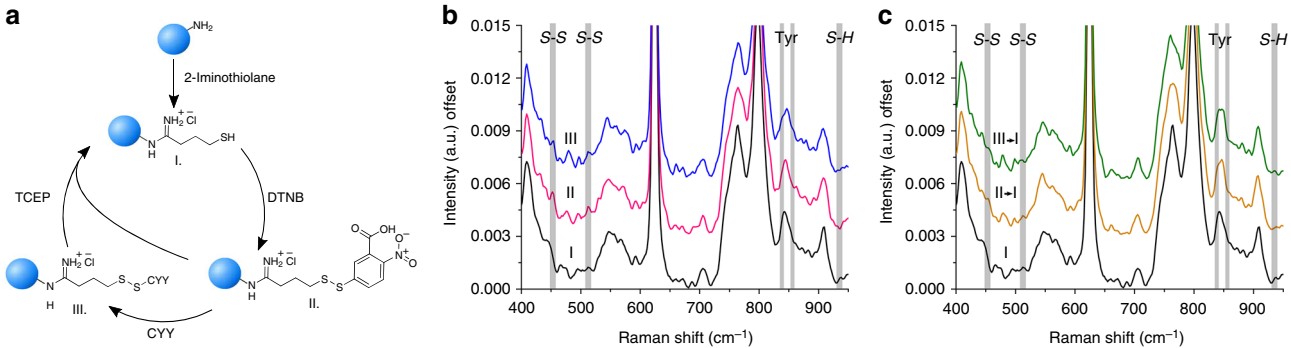

**Fig. 4** SPARTA functionalisation analysis of PS nanoparticles. **a** Schematic overview of the PS nanoparticle (blue sphere) functionalisation path. Amine functionalised particles are treated with 2-iminothiolane (Traut's reagent) leading to sulfhydryl functionalised particles (I). Addition of 5,5′-dithio-bis-(2-nitrobenzoic acid) (DTNB) leads to disulphide bond formation (II), which can be exchanged by the tripeptide consisting of cysteine-tyrosine-tyrosine (CYY) (III). Addition of Tris(2-carboxyethyl) phosphine (TCEP) to a disulphide containing particle returns the sulfhydryl functionalisation. **b** Averaged offset Raman spectra of trapped PS particles with sulfhydryl functionalisation (black, $n = 201$), disulphide and nitrobenzoic acid (magenta, $n = 119$) and disulphide and CYY functionalised particles (blue, $n = 122$). Bands indicate the characteristic *S–S* stretch (452, 512 cm$^{-1}$), tyrosine ring breathing (*C = C* 840, 860 cm$^{-1}$) and *S-H* bend (936 cm$^{-1}$) vibrations. **c** Averaged offset Raman spectra of trapped PS particles with sulfhydryl functionalisation (black, $n = 201$) recovery of sulfhydryl after addition of TCEP to disulphide and nitrobenzoic acid functionalisation (orange, $n = 119$), and to CYY functionalisation (green, $n = 115$)

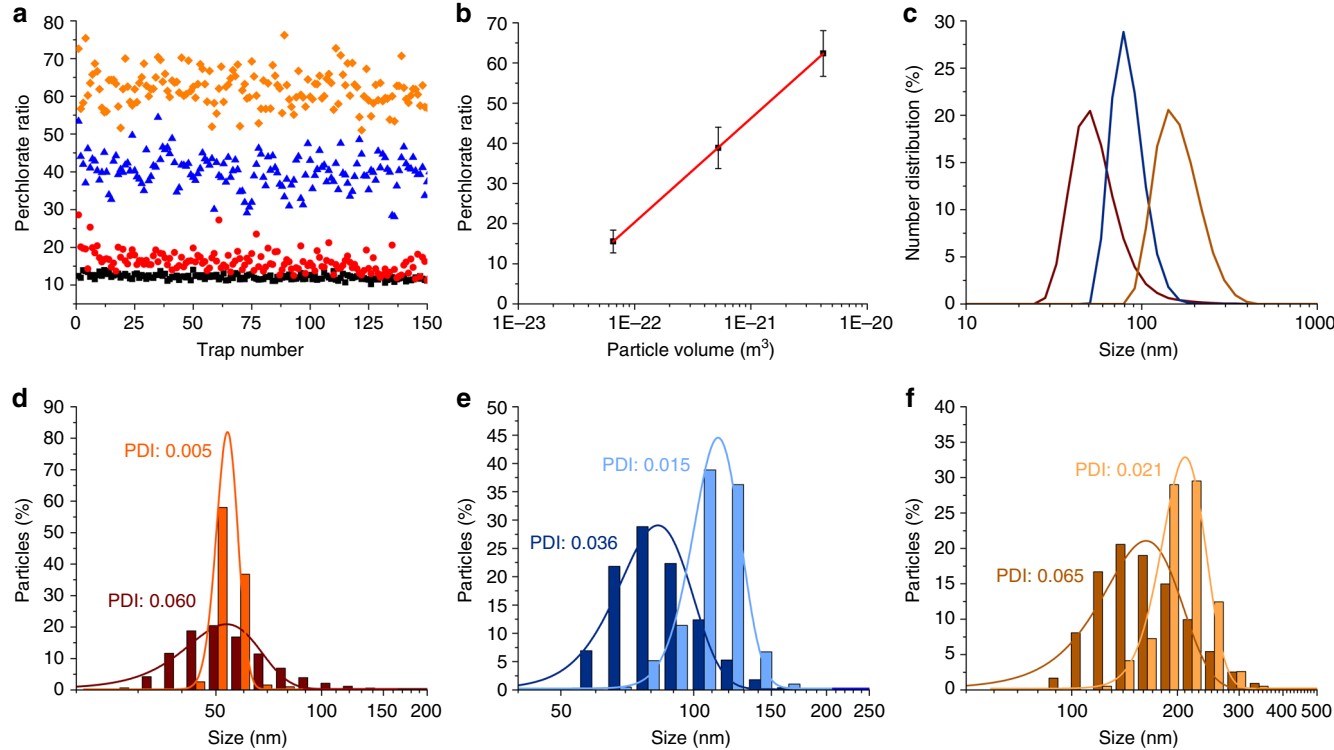

Fig. 5 SPARTA solution marker mediated sizing analysis. **a** Scatterplot of the perchlorate ratio for 150 traps of 50 nm (red circles), 100 nm (blue triangles) and 200 nm (orange diamonds) PS particles and PBS background (black squares) containing 50 mM sodium perchlorate in the solution. **b** Average perchlorate ratio (mean ± s.d.) versus the particle volume, a linear fit ($R^2 = 0.99$) is shown in red. **c** DLS number distributions of PS particles (mean ± s.d.), 50 nm (red, 53.6 ± 13.1 nm), 100 nm (blue, 83.0 ± 15.7 nm) and 200 nm (orange, 162.8 ± 41.5 nm). **d–f** Histogram of particle sizes measured by DLS (dark) and calculated from the calibration curve from the Raman spectra (light) including respective Gaussian fits of **d** 50 nm, **e** 100 nm and **f** 200 nm PS particles

tions of the particles (Fig. 5c) to yield a histogram of the size distribution (Fig. 5d–f). The main means of comparison between the DLS and SPARTA size measurements is the distribution broadness. This can be characterised by the polydispersity index (PDI), which is the square of the quotient of the standard deviation (s.d.) and the mean. For each particle size, the PDI obtained by the SPARTA solution marker-mediated sizing analysis was significantly lower than measured by DLS, demonstrating the ability of SPARTA for accurate single particle sizing analysis.

**On-line dynamic reaction monitoring.** The third mode of the SPARTA platform is the capability for on-line dynamic reaction monitoring, either on a single particle during the timescale of the reaction, or via continuous sampling of different single particles from the population. This allows the distinction between two different reaction scenarios, where the reaction proceeds uniformly throughout the whole population simultaneously, or where the reaction is initiated on different particles sequentially.

We used a model system to investigate the dynamics of the copper catalysed azide-alkyne cycloaddition (CuAAC) reaction, a type of click reaction which is frequently employed for nanoparticle functionalisations[35,36]. We obtained PS particles with an alkyne functionality via the EDC-NHS mediated coupling of propargyl amine to carboxylated PS particles with an average size of 200 nm. This alkyne was subsequently reacted, in the presence of a copper catalyst, with an azide-containing moiety resulting in the formation of a triazole ring (Fig. 6a). We verified the successful alkyne functionalisation of the PS particles by SPARTA (Fig. 6b), showing a characteristic Raman peak at 2129 cm$^{-1}$. The CuAAC reaction was subsequently conducted while trapping single particles sequentially or holding a single particle

continuously in the trap. Both the alkyne and azide signals of the reactants show a clear decrease over time, with the peak intensity of the triazole product increasing in an inverse trend (Fig. 6c). We observed reaction completion after ~8 min. Adapting the system to instead hold a single particle in the trap and monitor the spectral changes continuously for the duration of the reaction resulted in a similar trend of reaction (Fig. 6d), albeit showing reaction completion within 2 min, taking into account an additional lead time between activation of the catalyst and acquisition of the first spectrum of ~30–60 s. In addition, to verify the successful reaction on the alkyne functionalised particles, we monitored the reaction of 3-Azido-7-hydroxycoumarin with the particles by UV-Vis fluorescence, as the triazole product of 3-Azido-7-hydroxycoumarin results in a characteristic fluorescent emission (Absorption/Emission = 404/477 nm). In the presence of the dye and reaction conditions, the fluorescence increased gradually and starts to level off within 30 min (Supplementary Fig. 6).

## Discussion

The SPARTA platform enables concurrent label-free multi-parameter, non-destructive characterisation of particles smaller than the diffraction limit. It combines the efficiency of optical trapping with the established sensitivity of Raman spectroscopy for measuring composition and functionalisation at high-throughput via automation. We have demonstrated efficient trapping and spectral analysis for multiple nanoparticle formulations including different liposome and polymersome compositions, and serial chemical functionalisation of PS particles. Simultaneously, we were able for the first time to evaluate the particle size and distribution based solely on Raman scattering of single particles, through addition of perchlorate as an example

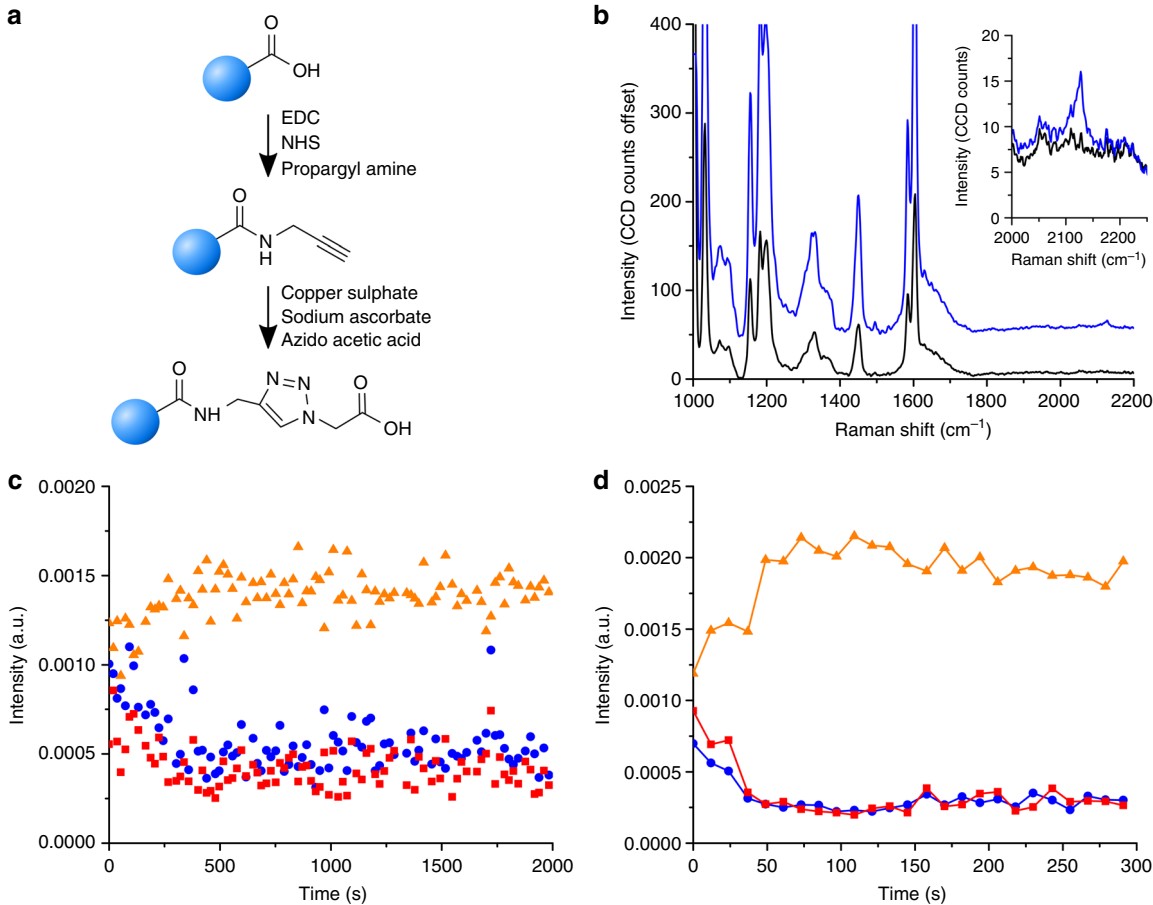

**Fig. 6** SPARTA on-line dynamic reaction monitoring. **a** Schematic overview of the functionalisation of PS nanoparticles with an alkyne moiety via EDC-NHS coupling with propargyl amine and subsequent CuAAC reaction with azido acetic acid to form the triazole product. **b** Offset averaged Raman spectra of trapped PS particles with carboxylic acid functionalisation (black, $n = 100$) and after alkyne functionalisation (blue, $n = 100$). Inset showing a close-up of the alkyne peak region, with a clear alkyne peak visible after functionalisation with a maximum at 2129 cm$^{-1}$. **c** Dynamic trace over time of the alkyne peak intensity (blue spheres, 2129 cm$^{-1}$), azide peak intensity (red squares, 2116 cm$^{-1}$) and triazole product (yellow triangles, 1331 cm$^{-1}$) for single particle trapped population dynamics and **d** single particle dynamics

solution marker. Furthermore, we have demonstrated the temporal analysis capabilities of this platform by monitoring a copper catalysed click reaction as it occurs on the surface of either a population sample or an individual nanoparticle. The automated evaluation of large samples of these particles can provide new insights into the heterogeneity of nanoparticle systems, investigate subpopulations and monitor dynamic composition changes for myriad sample designs.

The SPARTA platform represents the first automated system to study chemical composition, functionalisation and size for nanoparticles within a single modality with the label-free chemical and temporal resolution to track on-particle dynamic processes. The SPARTA platform is ideally suited to polymer and lipid systems but can be extended to a wide range of particle formulations with unique vibrational spectra. The technology described here is limited by fundamental properties; it is particularly well suited to samples with low turbidity to generate the particle trapping force gradient, compositions with non-overlapping spectral features, dynamic processes that occur over the course of minutes up to a few hours, and sample concentrations low enough such that no low-level background is contributed to the trapped particle measurement. Furthermore, each target particle system will have unique optical properties that must be further assessed for potential photothermal or other

photo-induced processes that can occur during trapping. Each of these considerations can be mitigated by careful selection and preparation of nanoparticle systems for evaluation.

The deployment of the SPARTA platform for complex nanoparticles provides multiplexed information regarding the intricate composition, size and dynamic processes of these systems and to improve fundamental understanding. Indeed, the SPARTA platform provides, in an automated fashion, feature-rich Raman spectra from hundreds of individually trapped particles in a single experiment. In summary, the complementary information obtained from the multifaceted SPARTA platform, regarding particle composition, functionalisation, size and dynamics, has enormous potential to critically impact fields including drug development and delivery, materials science and cellular biology.

## Methods

**SPARTA Raman micro-spectroscopy system**. Confocal Raman spectral acquisition was performed on a Raman micro-spectroscope (alpha300R +, WITec, Ulm, Germany). The light source used was a 785 nm laser (Toptica XTRA II) with a $63 \times /1.0$ NA water immersion microscope objective lens (W Plan-Apochromat, Zeiss, Oberkochen, Germany). The scattered light was collected via a 100 μm fibre with a 600 groove mm$^{-1}$ grating spectrograph (UHTS 300, WITec, Ulm, Germany) and spectra were acquired using a thermoelectrically cooled back-illuminated CCD camera (iDus DU401-DD, Andor, Belfast, UK) with a spectral resolution of 3 cm$^{-1}$ and 85 mW laser power at the sample. Laser control was performed remotely via a

serial connection and custom MATLAB (2016b, The Mathworks, MA, USA) scripts.

**SPARTA standard sample preparation.** For SPARTA analysis typically 200 µl of particle solution was required, of which approximately half was routinely recovered, depending on the measurement time. Ideal particle concentrations were determined to be between $1 \times 10^{10}$–$1 \times 10^{12}$ particles per millilitre or ~0.1–0.01% solids for PS particles. Sample solutions were placed on a 22 mm coverslip, affixed to a standard microscopy slide with a drop of phosphate buffered saline (PBS). The sample was placed under the water immersion objective for measurement.

**SPARTA standard data analysis.** The following preprocessing procedure was applied to all spectra acquired with the SPARTA platform. The spectral centre during standard acquisition was set at $1000\ cm^{-1}$ and the raw data was truncated to the range of $350$–$1825\ cm^{-1}$ to omit the excitation signal. For the measurements including the alkyne modification, the spectra centre was shifted to $1500\ cm^{-1}$ resulting in a measurement range of $606$–$2254\ cm^{-1}$. An automated script based on peak amplitude and 2nd derivative was employed for cosmic spike removal, followed by a manual visual check. Spectral background was subtracted via curve fitting (Whittaker filter, $\lambda = 100{,}000$) and noise smoothed using a Savitzky-Golay filter (3 points, first order). The resulting spectra were normalised via area under the curve, except for the SPARTA sizing analysis as this is incorporated via the perchlorate ratio calculation. Subsequent statistical analyses (hierarchical cluster analysis, PCA, multivariate curve regression, partial least-squares discriminant analysis) were implemented using PLS Toolbox (Eigenvector Research, Inc. WA, USA).

**DPPC and d-DPPC liposomes preparation.** Liposomes were prepared according to the following standard procedure. $5\ mg\ ml^{-1}$ stock solutions of lipid DPPC and 1,2-dipalmitoyl-d62-sn-glycero-3-phosphocholine (DPPC-d62, referred here to as d-DPPC), (Avanti Polar Lipids Inc. AL, USA) and cholesterol (Sigma-Aldrich, UK) were made in chloroform and stored at $-20\,°C$ under Argon prior to use. For DPPC liposomes a lipid film was made by adding 500 µl DPPC stock and 43 µl of cholesterol stock solution in a 10 ml round bottom flask resulting in a mol ratio of 85 : 15 mol % DPPC : cholesterol. For d-DPPC liposomes 250 µl DPPC stock and 250 µl d-DPPC stock was used, resulting in a ratio of 42.5 : 42.5 : 15 mol % // DPPC : d-DPPC : cholesterol. The chloroform was evaporated under nitrogen flow to form a thin lipid film. Lipid films were lyophilised overnight in a freeze dryer (Labconco, MO, USA) prior to rehydration. The films were hydrated with 1 ml PBS, shaken for 1 min and sonicated for 1 min. The solutions were then extruded 31 times through a polycarbonate membrane (Avanti Polar Lipids Inc. AL, USA) with a mesh size of 200 nm at 60 °C. Liposome size distribution and particle concentration were determined via NTA.

**DPPC liposome photostability.** A 100 times dilution in PBS of the liposome solution as described above was made and a liposome was trapped and held in the laser beam (785 nm, 85 mW) of the SPARTA system. Spectra of 10 s integration time were acquired with a 1 s interval between measurements. A spectral series of 5 min was identified as having one particle stable in the trap consisting of 26 consecutive measurements. The mean and standard deviation of the raw truncated spectra were determined with OriginPro 2016 (OriginLab Corporation, MA, USA). A surface plot of the raw truncated data was obtained with MATLAB (2016b).

**Polymersome preparation.** PMOXA-b-PDMS-b-PMOXA polymersomes (denoted further as ABA) were prepared from the poly(2-methyloxazoline-b-dimethyl-siloxane-b-2-methyloxazoline) ($M_n \cdot 10^3 = 0.5$-b-4.8-b-0.5) triblock copolymer (P18140D-MOXZDMSMOXZ, Polymer Source Inc., Quebec, Canada). 1 ml of a $6\ mg\ ml^{-1}$ stock solution of the triblock copolymer in ethanol was added to a 5 ml round bottom flask. ABA-heparin polymersomes were made by mixing in 25 wt % PDMS-b-heparin (Mw = 5kDa-b-11kDa) block copolymer synthesised as previously described by Najer et al.[30] Briefly, commercial heparin sodium salt (15 kDa, Merck, KGaA, Germany) was ion-exchanged to tetrabutylammonium salt and reacted in DCM with an excess of commercial diamino-PDMS (5 kDa, ABCR GmbH, Germany) via reductive amination using 2-picoline borane (Sigma-Aldrich GmbH, Germany) as reducing agent. The reaction was stirred for 7 days at RT with two more additions of 2-picoline borane on day three and five. The product was dried, washed in diethyl ether, dried, dissolved in ethanol, purified by repeated precipitation in cold diethyl ether and dried. 1 ml of $6\ mg\ ml^{-1}$ ABA stock was combined with 0.5 ml $4\ mg\ ml^{-1}$ stock of PDMS-b-heparin block copolymer in ethanol in a 5 ml round bottom flask. The polymer solutions were dried on a rotary evaporator at 50 °C and 20 mbar for ~15 min. Subsequently, the polymer films were rehydrated in 1.2 ml PBS for 72 h under vigorous stirring. The polymer solutions were filtered through a 0.45 µm syringe filter (Millex-HV 13 mm PVDF, Merck KGaA, Germany) and extruded 5 times through a polycarbonate membrane (Avanti Polar Lipids Inc. AL, USA) with a mesh size of 200 nm and subsequently 31 times through a polycarbonate membrane with a mesh size of 100 nm. The polymersomes were further purified by size exclusion chromatography (SEC) (10 × 300 mm column packed with Sepharose 2B (Sigma-Aldrich, UK)) in PBS,

collecting 1 ml fractions. Polymersome size distributions were analysed by DLS and NTA.

**Nanoparticle Tracking Analysis (NTA).** NTA (NS300, 532 nm laser, Malvern, UK) was performed by acquisition of 3 times 30 s videos of a 1 ml sample in PBS. The camera level was kept between 13 and 14, with a screen gain of 1 and detection threshold set at 5. The samples were diluted to within the optimum measurement range of $1 \times 10^8$–$1 \times 10^9$ particles per millilitre for measurement. The measurements were analysed using the Nanosight NTA 3.0 software (Malvern, UK, 2014).

**Dynamic light scattering (DLS).** DLS (ZEN3600 Zetasizer, Malvern, UK) was performed within disposable semi-micro cuvettes (Brand GMBH, Germany) with 400 µl solution, through acquisition and averaging of 3 measurements (each of 10–15 acquisitions) by NIBS at 173° scattering angle. The measurements were acquired using the Zetasizer Software v.7.02 (Malvern, UK, 2013). The number distributions were used to verify and compare particle size distributions.

**Preparation of Cysteine-Tyrosine-Tyrosine (CYY) tripeptide.** CYY tripeptide was synthesised by standard solid phase peptide synthesis using Fmoc protecting group chemistry on Rink-amide MBHA resin and protected Cysteine and Tyrosine amino acids (AGTC Bioproducts Ltd.). Briefly, Fmoc deprotection was performed with 20 v% piperidine in DMF for 10 min, followed by two washes with DMF and DCM. Amino acid couplings were carried out with Fmoc-protected amino acids (4 equivalents), HBTU (3.75 equivalents), and DIEA (6 equivalents) in DMF for 2 h and the process repeated as per the sequence. The peptide was cleaved from the resin and deprotected with 95% trifluoroacetic acid (TFA), 2.5% tri-isopropylsilane and 2.5% water for 4 h. The TFA was removed using rotary evaporation, and the peptide was precipitated and washed with cold diethyl ether 200 ml and 2 × 50 ml. For purification, the peptide was dissolved in a solution of 4.9% ACN in ultrapure water with 0.1% TFA and purified using reverse-phase preparative high-performance liquid chromatography (HPLC; Shimadzu, Japan) with a C18 Gemini 150 × 21.2 mm column (Phenomenex, CA, USA) with a 5 µm pore size and a 100 Å particle size. The mobile phase was ultrapure water containing 0.1% TFA @ 15 ml min$^{-1}$ and during the 15 min run the concentration of ACN containing 0.1% TFA in the mobile phase changed was 0% 0–3 min, 0–100% 3–12 min, 100 % 12–13 min and 0 % 13–15 min. The HPLC fractions were checked for the correct mass using Liquid Chromatography-MS (LCMS, Agilent, CA, USA) (Observed MW = 447.2, Predicted [CYY] H$^+$ = 447.16), and the pure peptide fractions were combined, rotary evaporated to remove ACN and lyophilised by freeze drying (Labconco, MO, USA).

**Serial functionalisation of polystyrene particles.** Amine functionalised 0.2 µm PS particles (Polybead Amino 0.20 µm, Polysciences Inc.) were further functionalised with 2-iminothiolane. A reaction buffer of 2 mmol EDTA in PBS was made and adjusted to pH 8 with 2 M NaOH, from which a $6\ mg\ ml^{-1}$ solution of 2-iminothiolane was prepared. For functionalisation 780 µl of reaction buffer was combined with 200 µl of 0.20 µm PS particles (2.6% solids (w/v)) and 20 µl 2-iminothiolane solution and left to react overnight at room temperature. This resulted in a 0.5% solids (w/v) solution of sulfhydryl functionalised particles, which were further diluted 10 times in PBS and purified. Unless otherwise stated, purification was performed by centrifugation for 10 min at 14,000 rcf, after which the pellet was redispersed in PBS. Redispersion was aided by vortexing for 30 s and ultrasonication for 1 min, obtaining a clear solution. After each purification step DLS measurements were performed to verify the absence of aggregation, prior to SPARTA. A minimum of 100 successful trapping spectra were acquired with the SPARTA experimental parameters set to 1 s iteration acquisition time, 10 s high SNR acquisition times and 1 s laser disabling time. The particles were further treated with 10 mg of 5,5-dithio-bis-(2-nitrobenzoic acid) (DTNB) forming a disulphide bond between the sulfhydryl functionalisation and the TNB anion. The particles were purified and SPARTA was performed to verify disulphide bond formation. The tripeptide functionalisation was obtained by treating the TNB functionalised particles with 2 mg of CYY ($M_w$ = 446.16 g mol$^{-1}$, 4.5 mM). SPARTA was performed after purification to observe the tripeptide functionalisation. To demonstrate the reversibility of the functionalisation, 100 µl of tris(2-carboxyethyl)phosphine (TCEP, 0.5 M, neutral pH Bond-Breaker™, ThermoFisher Scientific, UK) was added to the TNB functionalised particles, turning the solution bright yellow indicating cleavage of the disulphide bonds. Similarly the disulphide bonds between the particle and the tripeptide were cleaved. After purification the recovery of the sulfhydryl functionalisation was verified by SPARTA.

**Dynamic click reactions on polystyrene particles.** Carboxyl functionalised 0.2 µm PS particles (Polybead carboxylate 0.20 µm, Polysciences Inc.) were functionalised with propargyl amine using EDC-NHS coupling. Solutions were made of $20\ mg\ ml^{-1}$ 1-ethyl-3-(3-dimethylaminopropyl)carbodiimide and $20\ mg\ ml^{-1}$ n-hydroxysuccinimide in PBS and 40 µl of each was added to 200 µl of PS particles (2.6% solids) and 800 µl of PBS. The solution was shaken on a thermomixer at room temperature and 20 µl of neat propargylamine was added after 30 min. The reaction was allowed to proceed under continuous shaking overnight. The synthesis solution was diluted 10 times and purified. Purification was performed by centrifugation for

10 min at 14,000 rcf, after which the pellet was redispersed in PBS. Redispersion was aided by vortexing for 30 s and ultrasonication for 1 min. After purification, DLS traces were obtained to verify the absence of aggregation, prior to SPARTA. Solutions were made of 100 mM copper sulphate, 100 mM sodium ascorbate and 0.5 M potassium bicarbonate in PBS. Azido acetate was made by addition of a 1 M solution of sodium hydroxide in a molar equivalent to 2-azidoacetic acid (Sigma-Aldrich, UK). The population click reaction was carried out by formation of the triazole by addition of 2.88 µl 100 mM sodium ascorbate, 1.80 µl 100 mM copper sulphate and 0.5 µl neat azido acetic acid to 200 µl of alkyne functionalised PS particles, diluted 100 times in PBS. Droplets of a 0.5 M solution of $KHCO_3$ were applied to adjust the pH to 7. The single particle hold click reaction was performed with addition of 7.46 µl azido acetate (equalling 0.5 µl neat azido acetic acid), with no further pH adjustment necessary.

**Click reaction monitoring by UV-Vis analysis**. 3-Azido-7-hydroxycoumarin (Jena Bioscience GmbH, Germany) was used to validate whether a CuAAC reaction would occur on the alkyne functionalised PS nanoparticles. To do this the fluorescence of the resulting triazole product (Absorption/Emission = 404/477 nm) was monitored by UV-Vis spectroscopy. In a 96 well plate 200 µl of a 1000 times dilution of the purified alkyne functionalised PS particles in PBS was combined with 2.88 µl 100 mM sodium ascorbate, 1.80 µl 100 mM copper sulphate and 5 µl 3-Azido-7-hydroxycoumarin 1 µM in water. As controls, the measurements were performed at the same time with exclusion of 3-Azido-7-hydroxycoumarin or copper sulphate. The fluorescence was monitored over 30 min and measured at 15 s intervals.

**Code availability**. Code is available on request from rdm-enquiries@imperial.ac.uk subject to any restrictions related to IP filing.

## Data availability

Raw research data are available online at https://doi.org/10.5281/zenodo.1338335.

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

## Acknowledgements

J.P. and C.C.H. acknowledge funding from the NanoMed Marie Skłodowska-Curie ITN from the H2020 programme under grant number 676137. I.J.P. acknowledges support from the Whitaker International Program, Institute of International Education, United States of America. M.S.B. acknowledges support from H2020 through the Individual Marie Skłodowska-Curie Fellowship "IMAGINE" (701713). C.S.W. acknowledges support from the i-sense Engineering and Physical Sciences Research Council (EPSRC) IRC in Early Warning Sensing Systems for Infectious Diseases (EP/ K031953/1; www.i-sense.org.uk). A.Najer was supported by a Swiss National Science Foundation Early Postdoc Mobility Fellowship (P2BSP2_168751), which is gratefully acknowledged. U.K. acknowledges support from the Deutsche Forschungsgemeinschaft [KA 4370/1-1]. A. Nagelkerke acknowledges support from NIHR Imperial Biomedical Research Centre and the Institute of Cancer Research, London, through the joint Cancer Research Centre of Excellence (CRCE). M.M.S. acknowledges a Wellcome Trust Senior Investigator Award (098411/Z/12/Z).

## Author contributions

J.P. and I.J.P. conceived the study and designed the SPARTA system, J.P. performed all SPARTA measurements, I.J.P., C.C.H. and M.S.B. aided in Raman data analysis, C.S.W. performed peptide synthesis and with U.K. aided in particle functionalisation and

dynamics, A.Najer performed polymer synthesis and aided in polymersome preparation. U.K. aided in DLS measurements. A.Nagelkerke aided in study design and discussions. J.P. and I.J.P wrote the manuscript with input from all authors. M.M.S. contributed to study design, scientific discussions, revised the manuscript and supervised the project.

## Additional information

**Competing interests:** J.P., I.J.P. and M.M.S. have filed a patent application (1810010.7) and trademark (UK00003308379) covering the name 'SPARTA' and the techniques as described in the manuscript, including but not limited to the process of automated particle recognition and the analysis and the sizing of trapped particles mediated by a solution marker. The remaining authors declare no competing interests.

