## [Peer Review File · Nature Communications]

Reviewers' comments:

Reviewer #1 (Remarks to the Author):

Penders, Pence et al describe automated Raman trapping analysis (SPARTA) that can be used for real-time characterization of composition, chemical reactions, self-assembly, and other processes with particles as small as 50 nm. Of particular interest are the organic nanoparticles exemplified by liposomes, nanoshells, exosomes, etc. SPARTA technique is powerful and versatile. To some degree, I wonder, why it was not yet done in this fashion until now. I can immediately see its usefulness both for organic and inorganic nanoparticles in 50-200 nm range. I strongly support publication of this manuscript in NComm. The following comments should be addressed.

1. It would be useful to have in the main text the data of successful vs unsuccessful trapping.
2. I would also recommend to add more extensive benchmarking regarding photoinduced processes in the trap, for instance photoionization. Self-disproportionation (process opposite to their self-assembly) of 'soft' organic particles in high intensity laser beams can happen as well.
3. Figure 6b. The change of intensity of Raman peaks is indeed quite useful. I would also expect to see the appearance of new peaks or disappearance of old ones upon functionalization.

Reviewer #2 (Remarks to the Author):

In this work the authors combine existing techniques such as optical trapping and Raman spectroscopy in an automated platform. In this way the particle throughput is increased, allowing to reach statistically more significant information on particle composition and functionalization. In addition, using a perchlorate ion marker and a calibration procedure, the volume/size of particles can be determined. Also dynamics of reactions occurring on or in particles can be monitored on the timescale of minutes. The method and its three working modes are clearly explained and convincingly demonstrated.

I follow the authors that by using such a fast automated platform a new range of possibilities opens up. Also the method for particle sizing based on adding a perchlorate marker is novel. Therefore I believe that the reported results are novel and will appeal to a broad audience. As a result I recommend to accept the paper as it is.

I did want to note that there exists an automated platform based on electrokinetic trapping of (fluorescent) nano-particles as described for example in Wang & Moerner, 2015 (PNAS.1514027112), which is able to characterize the size of particles, particle conformation/interaction/functionalization, and obtain spectral information (from fluorescence). In some aspects there is a similarity, however no Raman analysis is carried out (as far as I know). Therefore I believe the novelty claim in this work is appropriate.

Reviewer #1

We thank the reviewer for saying they strongly support publication in Nature Communications. We have addressed the comments of the reviewer below.

1. It would be useful to have in the main text the data of successful vs unsuccessful trapping.

A comparison of the spectra obtained in a successful vs unsuccessful trap is made in Figure 2 ('IV. Failed trap' and 'V. Successful Trap') as the data presented are actual spectra obtained during measurements of polystyrene particles. During the developed automated trapping set-up, the unsuccessful spectra are not recorded and saved but the iterations are continued until the trapping is recognized above threshold (V.) after which a high SNR acquisition occurs (VI.) and is saved for further processing (VII.).

To clarify this direct comparison, a statement has been added to the manuscript indicating the opposing outcomes for trapping evaluations:

'The automated determination of a successful particle trap obviates the need to record data from iterations without sufficient signal relative to the threshold as can be seen by the comparison of the spectral lineshape for a successful polystyrene particle trap (Fig. 2a-V) versus a trapping iteration containing only the fluid medium (Fig. 2a-IV).'

We hope that this comparison between successful vs unsuccessful trapping is sufficient.

2. I would also recommend to add more extensive benchmarking regarding photoinduced processes in the trap, for instance photoionization. Self-disproportionation (process opposite to their self-assembly) of 'soft' organic particles in high intensity laser beams can happen as well.

We would like to thank the reviewer for this suggestion, as it is indeed important to show that no undesirable photodamage or photo-initiated reactions occur on the particles. To demonstrate this we have added to the SI, now Fig. S1. Here a DPPC liposome was continuously held in the trap for 5 min and spectra were acquired approximately every 10s. As can be seen in Fig. S1a the standard deviation around the raw mean spectrum of the DPPC liposomes is very low and does not indicate any appreciable changes over time. In addition in Fig. S1b a surface plot is shown where the spectrum can be seen over time, with no indications of photodamage or other laser induced effects on the particle visible, except for a light fluctuation in the overall intensities that can be attributed to slight fluctuations in the laser power.

We would also like to draw the reviewer's attention to the references 14 (Ajito *et al.*, 2001) and 27 (Cherney *et al.*, 2010) of the manuscript where the following assertions are made:

'The optical tweezers initially used a visible laser beam. Nowadays though a low-energy near-infrared (NIR) laser beam ranging from 700 to 1100-nm wavelength is widely used because it produces far fewer sample-damaging photochemical reactions than visible laser light. An NIR laser beam focused on a small particle using an objective lens traps the particle without damaging it.' (Ajito *et al.*, 2001)

'To prevent photodamage by electronic excitation, investigators commonly use deep red and near-infrared lasers for trapping.' (Cherney *et al.*, 2010)

The potential for photo-induced processes are certainly of interest with respect to the target particles; by utilizing the NIR excitation wavelength we have minimized the potential for single or two-photon absorption to cause electronic excitation that is characteristic of higher energy laser sources. Furthermore, for each of the particle systems investigated thus far, the strongest absorber at this wavelength is the fluid medium containing the particles and with continuous irradiation should result in less than 1 °C rise associated with the trapping and Raman measurement (Leitz *et al.*, 2002). Due to this fact, even small local heating changes on particles due to weak optical absorption, heat transfer should further minimize photothermal effects. This does not preclude the possibility of either photothermal or other photo-induced processes for other particle targets, which are

interesting points for further investigation by this group and others once this urgent proof of principle report has been published.

3. Figure 6b. The change of intensity of Raman peaks is indeed quite useful. I would also expect to see the appearance of new peaks or disappearance of old ones upon functionalization.

This is indeed what we aim to show in Fig. 6b, in particular in the inset, that upon functionalisation with propargyl amine the specific alkyne Raman peak (2129 cm^{-1}) appears in the spectrum of the trapped particles. To clarify this further we have changed the wording of the figure caption slightly:

'(b) Offset averaged Raman spectra of trapped PS particles with carboxylic acid functionalisation (black, $n = 100$) and after alkyne functionalisation (blue, $n = 100$). Inset showing a close-up of the alkyne peak region, with a clear alkyne peak visible after functionalisation with a maximum at 2129 cm^{-1} .'

Reviewer #2

We thank the reviewer for recommending acceptance of the paper as is.